# Fibrates: A Possible Treatment Option for Patients with Abdominal Aortic Aneurysm?

**DOI:** 10.3390/biom12010074

**Published:** 2022-01-05

**Authors:** Naofumi Amioka, Toru Miyoshi

**Affiliations:** Department of Cardiovascular Medicine, Graduate School of Medicine, Dentistry and Pharmaceutical Sciences, Okayama University, Okayama 700-8558, Japan; miyoshit@cc.okayama-u.ac.jp

**Keywords:** abdominal aortic aneurysm, fibrates, peroxisome proliferator-activated receptor alpha, macrophages, oxidative stress, matrix metalloproteinase

## Abstract

Abdominal aortic aneurysm (AAA) is a life-threatening disease; however, there is no established treatment for patients with AAA. Fibrates are agonists of peroxisome proliferator-activated receptor alpha (PPARα) that are widely used as therapeutic agents to treat patients with hypertriglyceridemia. They can regulate the pathogenesis of AAA in multiple ways, for example, by exerting anti-inflammatory and anti-oxidative effects and suppressing the expression of matrix metalloproteinases. Previously, basic and clinical studies have evaluated the effects of fenofibrate on AAA. In this paper, we summarize the results of these studies and discuss the problems associated with using fenofibrate as a therapeutic agent for patients with AAA. In addition, we discuss a new perspective on the regulation of AAA by PPARα agonists.

## 1. Introduction 

Abdominal aortic aneurysm (AAA) is characterized by the progressive dilation of the abdominal aorta that can lead to sudden death in a patient because of aortic rupture. Studies have reported that the primary pathogenesis underlying AAA development occurs as follows: (A) inflammatory cytokine production- and macrophage infiltration-associated inflammation [1,2], (B) oxidative stress [3,4], (C) vascular smooth muscle cell apoptosis accompanied by reduced elastin production [5,6], and (D) extracellular matrix degradation caused by activated matrix metalloproteinases (MMPs) in the vascular wall [7,8,9]. Although accumulated evidence has delineated the detailed pathophysiology of AAA development, an effective clinical treatment for patients with AAA remains to be established. 

Fibrates are agonists of peroxisome proliferator-activated receptor alpha (PPARα), which is a transcription factor that is mainly expressed in hepatocytes. Binding of fibrates to PPARα either activates or inhibits the genes involved in lipid metabolism [10]. In contrast, activation of PPARα can regulate the expression of genes involved in inflammation and oxidative stress [11,12,13]. Notably, PPARα is expressed in almost all the cells of the body, including macrophages, vascular smooth muscle cells, and endothelial cells [12,14,15,16,17,18]. Previously, studies have demonstrated that fibrates decrease the production of inflammatory cytokines, infiltration of monocytes, and expression of MMP genes [19,20,21,22]. Fibrates have also been implicated in decreasing the production of antioxidative enzymes, including superoxide dismutase and catalase, in the aortic wall [23]. Since the mechanism of action of fibrates is counteractive to the mechanism of AAA pathogenesis, fibrates are potentially valuable therapeutic agents for AAA treatment (Figure 1).

## 2. Discussion

Several basic and clinical studies have evaluated the protective effects of fenofibrate, the most common PPARα activator, on AAA. Golledge et al. (2010) reported that pre-administration of fenofibrate (100 mg/kg/day) to a hypercholesterolemic mouse model reduced angiotensin II (Ang II)-induced aortic expansion in the model. This reduction was associated with decreased expression of proinflammatory cytokine osteopontin (OPN) and macrophage infiltration in the aortic wall [24]. In addition, Krishna et al. (2012) reported that pre-administration of fenofibrate (100 mg/kg/day) significantly reduced suprarenal aortic dilatation induced by Ang II infusion in hypercholesterolemic mice; this was accompanied by a decrease in the abundance of macrophages, lymphocytes, and apoptotic cells in the aortic walls [25].

However, two randomized controlled trials that assessed the effects of fenofibrate on AAA in humans revealed differing results. In the FAME (Fenofibrate in the management of AbdoMinal aortic anEurysm) trial, patients scheduled to undergo open AAA repair (n = 43) were treated with fenofibrate (145 mg/day) or a placebo for at least 2 weeks before their surgeries [26]. In this trial, although the serum triglyceride (TG) levels were significantly reduced by the fenofibrate treatment, the concentration of OPN or the number of macrophages in the aortic tissue was not significantly different between the two groups. In the FAME-2 trial, 140 patients with AAA were enrolled and treated with fenofibrate (145 mg/day) or a placebo for 24 weeks. However, the fenofibrate treatment did not significantly reduce the serum concentration of OPN or the rate of AAA progression [27]. 

There are several possibilities as to why fenofibrate exhibits inconsistent protective effects against AAA in basic and clinical studies (Table 1). 

First, this discrepancy may be because the human pathology during the development of AAA is difficult to mimic in animal models. In humans, this development is affected by various factors, including genetics, ethnicity, age, sex, smoking habit, alcohol consumption, hypertension, hyperlipidemia, and renal function [28,29,30,31,32,33,34,35,36]. Therefore, it is extremely difficult to reproduce an environment that takes into account all these factors that contribute to AAA development in mouse models. Furthermore, small AAAs have a slower rate of progression than large AAAs. For instance, a meta-analysis reported that a 3.5 cm AAA takes approximately 6 years to grow to 5.5 cm [38]. Another meta-analysis demonstrated that to maintain the risk of an AAA rupture in male patients to under 1%, follow-ups are required at an estimated interval of 8.5 years for a 3.0 cm AAA and 17 months for a 5.0 cm AAA [39]. On the contrary, a preliminary study using ultrasonography demonstrated that AAAs in hypercholesterolemic mice progress rapidly within 1 week and usually rupture within 2 weeks following AAA induction [37]. Therefore, it is highly possible that the effects of fenofibrate on the rapidly progressing AAAs in mice cannot reflect the effects in humans. In addition, drug treatments in basic studies are usually provided either before or during the administration of the stimulants that induce AAA. However, these protocols only evaluate the favorable effects of the treatments during the acute phase of AAA formation. For example, previous studies that have assessed the impact of doxycycline on AAA development in murine models reported varying results for doxycycline administration before and after AAA induction [41,42,43]. Notably, mice were administered fenofibrate before AAA development, induced by Ang II, in both the basic studies we discussed previously [24,25]. Therefore, it is possible that fenofibrate does not regulate oxidative stress, inflammation, or MMP-mediated extracellular matrix degradation in vascular walls of already-formed AAA. 

Another possibility is that fenofibrate does not effectively regulate the pathogenesis of AAA in humans. This may be because the efficacy of a fibrate is clinically limited because of its dose-dependent side effects, such as liver damage and increased serum creatinine levels [44,45,46]. In addition, fenofibrate needs to be administered at a higher concentration to activate human PPARα than that needed to activate mouse PPARα [40]. This is a critical weakness of fenofibrate as a PPARα agonist. On the contrary, pemafibrate is a recently developed selective modulator of PPARα [47,48]. It has the potential to enhance PPARα activation even at low effective concentrations, and it has lower off-target side effects than fenofibrate [49,50]. A clinical study has demonstrated that pemafibrate (0.2 and 0.4 mg/day) is significantly more effective in lowering the serum TG levels than fenofibrate (106.6 mg/day). It also lowers the rates of adverse drug reactions in patients with elevated serum TG levels and decreases the serum levels of high-density lipoprotein cholesterol [51]. Furthermore, in basic research, pemafibrate (1 mg/kg/day) exhibited stronger effects in reducing the expression levels of vascular cell adhesion molecule 1, macrophage marker F4/80, monocyte chemoattractant protein 1, and interleukin-6. Notably, it ameliorated the development of plaque formation in the aortic walls of hypercholesterolemic mouse models compared to fenofibrate (250 mg/kg/day) [52]. Future studies are warranted to evaluate the efficacy of pemafibrate on AAA.

## 3. Conclusions

In this article, we describe the problems of reproducing the therapeutic effects of fibrates against AAA, from basic studies to clinical research. This is because fenofibrate can only be administered at limited concentrations in human subjects to activate PPARα; high concentrations of fenofibrate increase the risk of off-target effects. Moreover, it does not significantly attenuate the inflammation and dilatation of AAA. Thus, future basic and clinical studies focusing on the impact of pemafibrate to treat AAA are required. These studies will help investigate the possibility of using PPARα agonists as treatment options for AAA.

## Figures and Tables

**Figure 1 biomolecules-12-00074-f001:**
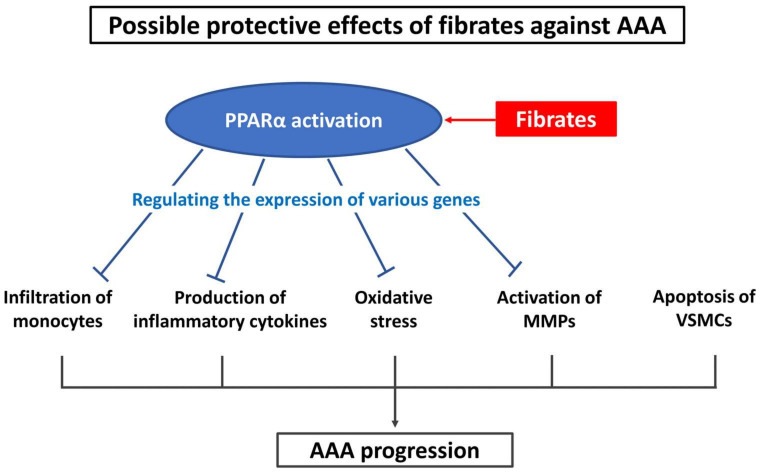
Possible protective effects of fibrates against AAA. AAA—abdominal aortic aneurysm; MMPs—matrix metalloproteinases; PPARα—peroxisome proliferator-activated receptor alpha; VSMCs—vascular smooth muscle cells.

**Table 1 biomolecules-12-00074-t001:** Difference in the progression and treatment of abdominal aortic aneurysm following fenofibrate administration in mice and humans.

	Basic Studies (Mice)	Clinical Studies (Humans)
Pathophysiological characteristics of AAA	Similar between each mouse	Substantially different between each patient (genetics, ethnicity, age, sex, smoking habit, alcohol consumption, hypertension, hyperlipidemia, and renal function) [28,29,30,31,32,33,34,35,36]
Rate of AAA progression	Rapid (days to weeks) [37]	Slow (months to years) [38,39]
Timepoint of fenofibrate administration	Prior to the time of AAA development (pre-treatment) [24,25]	Following AAA development (treatment) [26,27]
Effective concentration for activation of PPARα by fenofibrate	Lower [40]	Higher [40]

AAA—abdominal aortic aneurysm; PPARα—peroxisome proliferator-activated receptor alpha.

## Data Availability

Not applicable.

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
