# Peer review of "Fibrates: A Possible Treatment Option for Patients with Abdominal Aortic Aneurysm?"

_biomolecules, 2022, doi:10.3390/biom12010074_

Round 1

Reviewer 1 Report

This is an interesting commentary, highlighting the possible role of fenobibrate and pemafibrate as a treatment for AAA. The commentary is well structured and gives an overview of possible pathophysiologic mechanisms. 

One minor comment: line 127-18 The assumption that pemafibrate is a promising therapeutic agent against AAA is not supported at the moment, since there is a lack of even preliminary evidence in humans. The authors should omit this sentence or at least rephrase.  

Author Response

Thank you very much for the important suggestion. In accordance with your suggestion, we have rephrased the corresponding part in the revised manuscript.

In addition, we have got the revised manuscript proofread for English language and style.

We have deleted the following sentences: (lines 127-)

Thus, the high efficacy of pemafibrate in suppressing vascular inflammation and atherosclerosis progression, without increasing the risk of off-target effects, makes it a promising therapeutic agent against AAA. Remarkably, it can potentially overcome the limitations that fenofibrate possesses in reproducing its therapeutic effects against AAA from animal models to the clinical scenario.

We have added the following sentence (highlighted in red font in the revised manuscript):

Future studies are warranted to evaluate the efficacy of pemafibrate against AAA.

Reviewer 2 Report

This is an interesting approach on the discrepancy between lab and clinical studies on the possible role of vibrates in reducing AAA-related inflammation. The paper is well written and concise. I only have one comment whether there are other vibrates besides  fenofibrate or pemafibrate that may apply for AAA management. If yes please refer.

Author Response

Thank you very much for the constructive comment. To our knowledge, there are no fibrates besides fenofibrate and pemafibrate that may be used for the management of AAA.

We would also like to inform you that we have got the manuscript proofread for English language and style.